# *Candida auris*: An Overview of How to Screen, Detect, Test and Control This Emerging Pathogen

**DOI:** 10.3390/antibiotics9110778

**Published:** 2020-11-05

**Authors:** Teresa Fasciana, Andrea Cortegiani, Mariachiara Ippolito, Antonino Giarratano, Orazia Di Quattro, Dario Lipari, Domenico Graceffa, Anna Giammanco

**Affiliations:** 1Department of Health Promotion, Mother and Child Care, Internal Medicine and Medical Specialities, University of Palermo, 90127 Palermo, Italy; anna.giammanco@unipa.it; 2Department of Surgical, Oncological and Oral Science (Di.Chir.On.S.), University of Palermo. Department of Anesthesia Intensive Care and Emergency, Policlinico “Paolo Giaccone”, 90127 Palermo, Italy; andrea.cortegiani@unipa.it (A.C.); ippolito.mariachiara@gmail.com (M.I.); antonino.giarratano@unipa.it (A.G.); 3Laboratory of Microbiology, Azienda Ospedaliera Ospedali Riuniti “Villa Sofia-V. Cervello”, 90127 Palermo, Italy; orazia.diquattro@villasofia.it; 4U.O.C. of Microbiology, Virology and Parassitology, A.O.U.P. “Paolo Giaccone”, 90127 Palermo, Italy; dariolipari79@gmail.com (D.L.); dome36g@hotmail.it (D.G.)

**Keywords:** *Candida auris*, *Candida auris* identification, screening, antifungal resistance testing

## Abstract

The multidrug-resistant yeast *Candida auris* is associated with invasive infections in critically ill patients and has been isolated in different countries worldwide. Ease of spread, prolonged persistence in the environment and antifungal drug resistance pose a significant concern for the prevention of transmission and management of patients with *C. auris* infections. Early and correct identification of patients colonized with *C. auris* is critical in containing its spread. However, this may be complicated by *C. auris* strains being misidentified as other phylogenetically related pathogens. In this review, we offer a brief overview highlighting some of the critical aspects of sample collection, laboratory culture-dependent and independent identification and the susceptibility profile of *C. auris*.

## 1. Introduction

*Candida auris*, the yeast pathogen firstly isolated from a Japanese patient’s external ear canal in 2009, has been involved in invasive healthcare-associated outbreaks and sporadic cases reported in various countries worldwide [1,2,3,4,5,6]. Three hundred forty-nine cases of *C. auris* were reported solely in the European Union between January 2018 and May 2019: most of these (73.6%) were colonizations and, among the infections, 24.1% were bloodstream infections [7].

This new microorganism has a severe impact on public health, not only because it is often multidrug-resistant, rapidly spreads among patients and persistently colonizes the skin and nosocomial surfaces, but also because it is often difficult to be correctly identified [8].

The European Centre for Disease Prevention and Control (ECDC) *C. auris* survey collaborative group reported that during 2013–17 several European countries lacked laboratory capacity and/or information on the incidence of cases at a national level [9]. Even if laboratory facilities and awareness seem to have improved since 2018, seven EU/EEA countries do not have a reference laboratory in their country yet [7]. Therefore, cases of *C. auris* may still be misidentified or unidentified and spread in healthcare settings. 

Misidentification of *C. auris* with other yeasts (e.g., *C. haemulonii, C. famata, C. guilliermondii, C. lusitaniae, C. parapsilosis*) may occur due to the use of standard biochemical methods and commercially available tests. In fact, its correct identification at the species level requires, as we will describe in this review, more advanced techniques, i.e., DNA sequencing or matrix-assisted laser desorption/ionization time-of-flight (MALDI-TOF), or both if the microorganism is isolated from sterile and nonsterile body sites, as it is now known that asymptomatic colonization represents a risk for *C. auris* transmission [10,11,12].

Misidentification may lead to a delay in the application of infection prevention principles. This delay is especially severe when considering the speed and ease of transmission of this yeast. *C. auris* can be transmitted by contact not only with colonized patients or healthcare personnel but also with contaminated surfaces from which this yeast is difficult to eradicate. In fact, its ability to switch into an aggregative form allows *C. auris* to persist and survive even under the effects of physical and chemical detergents [10]. Experiments conducted by Welsh and colleagues have documented the isolation of metabolically active viable but nonculturable (VBNC) cells of *C. auris* for at least four weeks, with viable colonies recovered by culture for at least two weeks, providing a first estimate of the environmental survival time of this species [13].

As previously mentioned, *C. auris* is often associated with antifungal drug resistance, which limits treatment options. Furthermore, the relationship between minimal inhibitory concentration (MIC) values and clinical outcomes is still not fully understood, resulting in a lack of consensus on the susceptibility breakpoints for *C. auris*.

Taking all this into consideration, it is clear how necessary it is for clinical laboratories to have a clear picture of how to quickly and correctly identify *C. auris*. This may lead to the implementation of infection prevention and clinical management strategies and help avoid the costs and risks of outbreaks weighing especially on immunocompromised and hospitalized patients with severe underlying disorders. The purpose of this review is to offer a simplified yet updated report on the available *C. auris* screening, identification and antifungal susceptibility determination methods while focusing on screening and control measures.

## 2. *C. auris* Screening

High-quality sampling procedures are needed to assess *C. auris* colonization [14]. For this purpose, sampling should be done using adequate equipment and transport systems (e.g., rayon tip swabs or nylon-flocked swabs) and screening should be done using a composite swab of the patient’s bilateral axillae and groin, as these are the most consistent sites of colonization. However, patients with *C. auris* infections may be often colonized at multiple body sites, such as in the nares, external ear canals, oropharynx, urine, wounds, vagina, rectum and catheter exit sites [11]. If topical antiseptic agents are administered, 48 h should elapse prior to sampling [11]. In the case of negative screens for *C. auris*, sampling should be repeated within seven days and before discontinuing any infection prevention and control measures. Then, it is recommended to postpone sampling procedures until three months later to reassess patient colonization or infection [15]. See Figure 1.

## 3. *C. auris* Identification

### 3.1. Phenotypic Methods and Distinctive Characteristics

The morphology of *C. auris* may resemble other more common *Candida* spp., thus making in vitro evaluation of colonies appearance impossible as the only laboratory identification method [16]. *C. auris* forms smooth and white/cream colonies on Sabouraund dextrose/glucose agar. In contrast, when growing on commercial chromogenic Candida agar medium, it forms pink, beige, pale rose or red colonies, which may be difficult to distinguish from *C. glabrata* [16].

Supplementation of commercial chromogenic media with Pal (sunflower seed extract) agar was reported to be useful, together with higher temperature, for the differentiation of *C. auris* from the *C. haemulonii* complex [17].

Recently, Borman and colleagues have also described a chromogenic agar, CHROMagar Candida Plus, for the specific identification of *C. auris* isolates. On this medium, *C. auris* colonies appear pale cream but present a distinctive blue halo. Of note, among over 50 different species of *Candida* spp and related genera that were cultured in parallel, only *Candida diddensiae* gave a similar appearance [18].

On microscopic examination, *C. auris* is characterized by oval or elongated yeast cells capable of existing as aggregate or nonaggregate cells. *C. auris* rarely forms pseudohyphae, depending on specific conditions of the media, such as the presence of NaCl [19]. *C. auris* shows multiple cellular morphologies when cultivated at 25, 37, and 40 °C using different growth media [20]. On Lee’s glucose and Lee’s GlcNAc media, *C. auris* cells exhibit an oval shape at 25 and 37 °C, and a relatively round shape at 40 °C. On Spider and agar plus serum media, cells are round and relatively small. *C. auris* cells are round on regular Yeast Extract–Peptone–Dextrose (YPD) Medium. At the same time, they take on an elongated shape on YPD plus 10% NaCl. Elongated cells of *C. auris* resemble opaque cells of *C. albicans* in shape. A small percentage of highly elongated and pseudohyphal-like cells are observed when grown on YPD plus 10% NaCl. Multiple nuclei are observed with DAPI staining in elongated cells [21]. However, no septin/chitin rings are observed between conjoint cells when Calcofluor white staining is applied [22].

Morphological diversity is a key virulence factor of *Candida* spp. [23,24]. However, it has been demonstrated that the pathways governing yeast-to-filament transition during morphogenesis and the related signals are different in *C. albicans* and *C. auris* [25]. Finally, *C. auris* can form biofilm, another virulent trait, which seems to be related to the type and phenotypic behavior of the isolates, as sessile and planktonic phenotypes were found to be associated with colonizing and clinical isolates, respectively [26].

### 3.2. Biochemical Methods

Most commercial biochemical tests commonly used may misidentify *C. auris*, as reported in a list of the most used methodologies for the identification of *Candida* species compiled and updated by the Centre for Disease Prevention and Control (CDC) (Table 1) [12].

*C. auris* may be misidentified as *Candida haemulonii, Candida sake, Rhodotorula glutinis* or other Candida species by these conventional biochemical systems because it is not always present in databases or because of the overlapping biochemical profiles [27]. In the case of the listed identification/instrument combinations, it is recommended to use the practical identification algorithms available in the CDC website [12].

It has been reported that VITEX 2 XL (bioMérieux version 8.01) is capable of detecting *C. auris* with high reliability. However, Ambaraghassi and colleagues showed that this software precisely identifies isolates from the South American clade, but had limited power to correctly identify *C. auris* from the African and East Asian clades [28]. Hence, all isolates identified using this system, as *C. auris, C. famata*, and species in the *C. haemulonii* complex should be confirmed by MALDI-TOF or DNA sequencing. 

Therefore, the methods listed above are currently not precise enough to identify this microorganism. Moreover, the slow turnaround time of enrichment cultures could be a significant limiting factor [29].

### 3.3. Molecular Methods

Both culture- and non-culture-based molecular methods for the detection of *C. auris* may be used [30,31,32,33,34,35,36,37,38,39,40,41,42,43,44] as summarized in Table 2.

Culture-based molecular methods rely on the yeast growth conditions (10% NaCl and 40 °C) and on its use of dulcitol as a carbon source to enrich *C. auris* over other Candida species [30,42]. Then, sequencing of the D1−D2 region of the 28S ribosomal DNA (rDNA) or of the internal transcribed region (ITS), or matrix-assisted laser desorption/ionization time-of-flight (MALDI-TOF) (to which we will dedicate paragraph 3.4.) can be used to identify this yeast rapidly [29]. It must be said that sequencing is not a common choice for routine identification due to its technical, timing and cost burden [45]. Culture-independent methods include PCR and/or real-time PCR (i.e., TaqMan quantitative PCR (qPCR) and SYBR green qPCR), T2 magnetic resonance assay and loop-mediated isothermal amplification (LAMP) [32,33,41,42]. All these methods exhibited clinical sensitivity (detection limits in the range of 1–10 CFU/reaction) and a specificity around 90%.

Moreover, they were confirmed by testing on large yeast panels and are time-saving as they reduce the overall turnaround time from days to working hours, allowing the rapid identification of colonized patients [29,32,33,37,40,41,46]. Lima and colleagues set up a TaqMan-based real-time PCR assay, which showed to be highly sensitive (PCR linearity of over 5 orders of magnitude with 99% efficiency), had a limit of detection (LoD) of 1 CFU (Ct = 34.3 ± 0.5) and had high specificity [43]. The use of molecular methods from swabs is also simple: commonly 100–200 µL of the total 1 mL of swab medium is used for DNA isolation, while the residual volume should be stored until results are obtained [29].

Currently, a T2 magnetic resonance assay (T2 Biosystems, United States) is available for the rapid (<3 h) and accurate and sensitive (1–3 CFU/mL) detection of specific Candida species (*C. albicans/C. tropicalis, C. parapsilosis* and *C. glabrata/C. krusei*) directly from blood specimens [41,42]. Moreover, with the addition of the *C. auris* panel, it detects this microorganism with a limit of detection < 5 CFU/mL in less than 5 h [41].

Kordalewska and colleagues developed two rapid, accurate, easy-to-perform molecular diagnostic assays based on real-time PCR to discriminate *C. auris* from other species. While the first assay identifies *C. auris* only, the second one—thanks to the use of SYBR Green (ThermoFisher Scientific, Waltham, Massachusetts) detection of amplicon and melting-point analysis—distinguishes between *C. auris, C. duobushaemulonii, C. haemulonii, and C. lusitaniae* [31]. Both PCR methods are highly reliable as they yielded 100% accuracy and concordance with the results of ITS region sequencing [31]. Of note, molecular methods may be costly (less than the MALDI-TOF) and detect DNA from both live and dead *C. auris* cells; in the case of a positive result from a nonsterile specimen (skin, environment), it could indicate past or present colonization with *C. auris*.

### 3.4. MALDI-TOF

MALDI-TOF mass spectrometry (MS) can reliably differentiate *C. auris* from colonies recovered in the enrichment broth, distinguishing them from other *Candida* spp. only if *C. auris* spectrum is included in the reference database [47]. Precise identification of *C. auris* is currently more likely with the FDA-approved MALDI-TOF Biotyper (Bruker-Daltonics), using their updated research use only (RUO) library (Versions 2014 5627 and more recent) or CA System library (Version Claim 4), or with the VITEK (MALDI-TOF) MS (bioMérieux) FDA-approved IVD v3.2 or the RUO Version 4.14 with the Saccharomycetaceae update [12,29].

Some authors reported higher accuracy and efficiency for the Bruker Biotyper over the Vitek 2 MS in the identification of *C. auris* isolates, even with an upgraded RUO [48].

In addition to these options, the CDC, in collaboration with Bruker, provides an online tool for the accurate classification of *C. auris* to the species level (https://www.cdc.gov/microbenet/index.html) [49]. Three different MALDI-TOF MS extraction protocols (on-plate, quick and extended tube) are available. Some authors recommended the use of the quick tube extraction method as it scored better confidence levels than the on-plate method, while not differing significantly from the extended method [50]. In spite of this, other authors indicated that the extended tube extraction for MALDI-TOF MS was not better than on-plate extraction [51]. Therefore, MALDI-TOF MS reliably and rapidly identifies *C. auris* isolates, compared to conventional identification methods. However, it represents a time-consuming method when applied to all colonies recovered from colonized patients and requires a substantial upfront investment [18].

## 4. Typing

Multilocus sequence typing (MLST), a proteomic analysis performed with MALDI-TOF MS, amplified fragment length polymorphism (AFLP) and whole genome sequencing (WGS), can be used to evaluate the genetic relatedness of *C. auris* isolates [47]. For MLST analysis, a set of four genetic loci, namely ITS, D1/D2, *RPB1* and *RPB2*, have been reported to be highly discriminatory for strain differentiation between *C. haemulonii* complex and *C. auris* [51].

For AFLP analysis, genomic DNA was subjected to a combined restriction–ligation procedure containing EcoRI and MseI restriction enzymes (New England Biolabs, Beverly, MA, USA) and complementary adaptors [52]. However, whole genome sequencing remains the gold standard for the determination of *C. auris* isolates clonality as sequence variation between individual strains within a clonal lineage is very small (typically 30–80 SNPs over a whole genome) [52,53,54,55].

Recently, de Groot and colleagues developed a short tandem repeat (STR) typing assay for *C. auris.* In concordance with WGS analysis, the authors identified five major different *C. auris* clusters (i.e., South American, South Asian, African, East Asian and Iranian) and most isolates differing by >30 SNPs were determined [56].

In a recent study, Vatanshenassan and colleagues compared different typing techniques (i.e., microsatellite typing, AFLP fingerprinting, ITS sequencing, MALDI-TOF MS and IR Biotyper FTIR spectroscopy) to evaluate their application in typing *C. auris* [57]. Results indicated microsatellite typing as the tool of choice for *C. auris* outbreak investigations because only this technique grouped the isolates into four main clusters, in accordance with WGS data. The other typing tools showed poor performances with the highest agreement between microsatellite typing and ITS sequencing with 45% similarity, followed by microsatellite typing and FTIR with 33% similarity. The lowest agreement was observed between FTIR spectroscopy, MALDI-TOF MS and ITS sequencing.

## 5. *C. auris* Resistance Profile and Antifungal Susceptibility Testing

### 5.1. C. auris Resistance Profile

One of the most concerning aspects of *C. auris* is its ability to develop resistance to all three of the main classes of antifungal drugs (azoles, echinocandins and polyenes), with severely limiting clinical and therapeutic management options [58,59,60,61]. Multidrug resistance (MDR), i.e., resistance to more than two antifungal classes, is observed in around 40% ≥ of *C. auris* isolates, with a small proportion (4%) exhibiting resistance to all classes of antifungals. Therefore, when in the presence of an unidentified yeast resistant to more than one antifungal drug, further testing for *C. auris* identification should be performed [8,14,16,55,62].

Recently, among 801 patients identified with *C. auris* in New York, three were found to have pan-resistant *C. auris* that had developed after treatment with antifungal medications, including echinocandins. All three patients had multiple comorbidities, but no recent domestic or foreign travel [63].

Zamith−Miranda and colleagues compared two clinical isolates of *C. auris* with distinct drug susceptibility profiles with a *C. albicans* reference strain using a multiomics approach. Their results highlighted that, despite the distinctive drug resistance profile, *C. auris* isolates were very similar. In contrast, their profile was different from that of *C. albicans*, both in terms of carbon utilization and in lipid and protein content, supporting a multifactorial mechanism of drug resistance [64].

In general, high levels of resistance to fluconazole are observed in *C. auris*, especially among isolates from India, where a study of 350 isolates showed around 90% minimal inhibitory concentrations (MICs) of fluconazole greater than 16 μg/mL, and South Africa [65,66,67,68,69,70]. *C. auris* can also, to a lesser extent, exhibit reduced susceptibility to other triazole antifungals (i.e., voriconazole, posaconazole, itraconazole and isavuconazole) [69]. As regards mechanisms of resistance to antifungals, we have already published a systematic review on this topic, so we will add, if available, only updated information [16]. *C. auris* commonly exhibits susceptibility to the polyene amphotericin B; however, geographical differences are found. For example, resistance to amphotericin B was detected in 30% of *C. auris* U.S. isolates [71]. The principal mechanism of amphotericin B resistance has not yet been identified in *C. auris*. Still, some authors related it to a reduction in ergosterol content in the cellular membrane [8]. Finally, the development of resistance to echinocandins—currently the first-line therapy drugs—has been observed in *C. auris* isolates from multiple geographic areas in patients initially treated with an echinocandin [8].

### 5.2. Antifungal Susceptibility Testing

Antifungal susceptibility testing (AFST) for *C. auris* can be performed using CLSI and EUCAST broth microdilution methods, as well as the E-test gradient diffusion method (bioMérieux), Sensititre YeastOne (Thermo Fisher Scientific) and Vitek-2 Yeast susceptibility system (bioMérieux). Nonetheless, these in vitro methodologies may present limitations, such as slow turnaround time (24 h after isolation) and, not lastly, the need of specific know-how to readout *C. auris* MIC values [29]. In Table 3 we have reported the principal AFST and their respective strengths and limitations.

Currently, no established susceptibility breakpoints are available for *C. auris*. Some epidemiologic cut-off values (ECVs) have been suggested by Arendrup and colleagues comparing the European Committee on Antimicrobial Susceptibility Testing (EUCAST) and the Clinical and Laboratory Standards Institute (CLSI) reference microdilution MICs, which appeared to have minor differences [65]. However, it must be said that ECVs are tentative breakpoints and isolates with MICs below or above the ECV should not be classified as susceptible or resistant, respectively, especially if considering that MIC distributions can vary substantially for *C. auris* isolates from different clades [66,72]. Moreover, the CDC provides guidance for *C. auris* MIC interpretation, based on information gathered for *Candida* spp. and several expert opinions (www.cdc.gov/fungal/candida-auris/c-auris-antifungal.html).

Talking about the need of specific know-how to readout *C. auris* MIC values, we must mention the possibility of running into the “eagle” effect or paradoxical growth effect of echinocandins, which is the reduced antifungal behavior at higher doses of the drug in vitro. It is important to underline that the eagle effect does not reflect the potential in vivo response of *C. auris* to echinocandins, as indicated in an invasive murine candidiasis model [73].

As regards automated systems for routine AFST, some authors observed a good (96.7%) agreement between VITEK 2 and CLSI methods in fluconazole AFST, although others have reported suboptimal performance for amphotericin B [74,75].

MALDI-TOF MS has also been increasingly used for AFST. Vella and colleagues have developed a rapid AFST assay based on MALDI-TOF MS analyzing changes in the MS profile spectra induced by antifungals after 3–6 h of incubation, firstly in caspofungin-resistant *C. albicans* and then in anidulafungin-resistant *C. glabrata* with known *FKS2* mutations, where they obtained less satisfactory results [76,77]. Recently, Vatanshenassa and colleagues used the MALDI Biotyper antibiotic susceptibility test rapid assay (MBT ASTRA) for the rapid detection of caspofungin-resistant *C. albicans* and *C. glabrata* and of *C. auris* [78,79]. The assay showed an accuracy of 100% on both agar plate and blood culture bottles and a sensitivity and specificity of 100% and 98% for anidulafungin and of 100% and 95.5% for micafungin, respectively. A categorical agreement of 98% and 96% was calculated for the two methods. For caspofungin, a sensitivity and specificity of 100% and 73% were found, respectively, with a categorical agreement of 82%. MBT ASTRA has the greatest potential to detect *C. auris* isolates nonsusceptible against echinocandin antifungals within 6 h, making it a promising candidate for AFST in clinical laboratories in the future [79].

In place of traditional methods, rapid molecular methods can be used to identify resistance-conferring mutations [29]. These methods can be used for high-throughput surveillance and provide important information quicker than standard methods, especially when performed with DNA isolated directly from swabs. For example, to simplify resistant *C. auris* screening, a duplex *ERG11* assay enabling detection of mutations at positions Y132 and K143, and a simplex *FKS1 HS1* assay enabling detection of mutations at position S639 were developed to identify known mutations related to resistance to azoles and echinocandins, respectively [80]. Results can be obtained in 2 h and have proven to be 100% concordant with DNA sequencing results [68]. Moreover, allele-specific probe technologies (e.g., Xpert MTB/RIF), molecular beacons, small stem-loop-structured DNA oligonucleotides, used with real-time PCR, have been developed for the detection and evaluation of antifungal resistance and enable distinguishing wild-type (WT) and mutant amplicons based on their melting temperature (T_m_). These assays are not only very reproducible but may also be updated each time a new mutation is detected [68].

## 6. Infection Control Recommendations

As previously said, *C. auris* has two characteristics that make it an epidemiologically important microorganism: the ability to spread rapidly and a multidrug resistance phenotype. For these reasons, its detection requires the implementation of basic infection control measures [62]. To contain transmission, once a case of *C. auris* is detected the ECDC recommends activating the screening of close contact patients, possibly extending contact tracing based on a case-by-case risk evaluation (e.g., type of patient and ward, level of colonization) [11]. Prompt notification to public health authorities and education of healthcare workers on the clinical impact of *C. auris* are also crucial. Meanwhile, point prevalence surveys should be run to identify colonized patients in hospital units where the index patient is or was present. It is also suggested to review patient records to determine any prior healthcare exposures, mainly overnight stays in healthcare facilities in the month prior to culture positivity [11].

When patients are moved to other healthcare facilities, notification of *C. auris* colonization/infection status is recommended [11]. For the same reason, screening of patients coming from geographical areas and healthcare facilities with a high incidence of *C. auris* infection/colonization at the moment of hospital admission is of utmost importance [11,80].

Infection control should include single room isolation or patient cohorting and dedicated nursing staff. Currently, there is no established decolonization protocol, so the above-described measures should be applied promptly. *C. auris* has proven susceptible to chlorhexidine in vitro in recent works; however, in a cohort of UK patients, despite daily chlorhexidine bathing, colonization with *C. auris* was not eradicated [6,81]. Further studies are needed to confirm the efficacy of chlorhexidine and other products for decolonization before they can be recommended for use. Moreover, it looks that the sessile/biofilm form of this yeast displays increased tolerance to clinically-relevant concentrations of chlorhexidine and hydrogen peroxide, with eradication achieved only using povidone-iodine [82].

However, it is known that reusable equipment may be a source of transmission of infection of *C. auris*. It is therefore crucial to have a cleaning protocol in place and strengthen the regular decontamination of the equipment and environment, especially high-touch areas [83].

## 7. Conclusions

The emerging pathogen *C. auris* has been associated with nosocomial outbreaks on five continents in the last ten years. Prevalence of infection is unpredictable due to misidentification and unreported cases. Proper microbiological identification, rigorous epidemiological surveillance, adequate treatment and prevention and containment strategies, combined with higher awareness on the side of physicians, microbiologists and healthcare workers, are indispensable to limit further spreading of this pathogen. Further research should investigate rapid and accurate laboratory identification methods and evaluate the clinical role of new therapeutic options to counteract *C. auris* antifungal resistance.

## Figures and Tables

**Figure 1 antibiotics-09-00778-f001:**
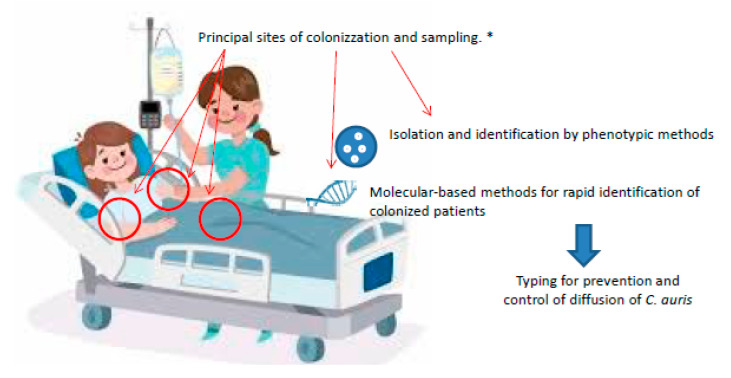
Screen, detect, test and control for control of diffusion of *C. auris.* * For patients with a strong hypothesis of *C. auris* infection, in the case of negative screens, sampling should be repeated within seven days and before discontinuing any infection prevention and control measures.

**Table 1 antibiotics-09-00778-t001:** Reported errors related to the identification of *C. auris* by commercially available biochemical tests.

Commercial Biochemical Systems	Misidentifies *C. auris* as
Vitek 2YST	*C. haemulonii* *Candida duobushaemulonii* *Candida* spp.
API 20C	*Rhodotorula glutinis* (characteristic red colour not present) *Candida sake* *Saccharomyces kluyveri* *Saccharomyces cerevisiae* *Candida* spp.
BD Phoenix	*C. haemulonii* *Candida catenulata* *Candida* spp.
MicroScan, Microscan Walkaway, MicroSan AutoScan	*C. famata* *Candida guilliermondii* *Candida lusitaniae* *Candida parapsilos* *Candida* spp. *R. rubra*
RapiID Yeast Plus	*C. parapsilosis* *Candida* spp.

**Table 2 antibiotics-09-00778-t002:** Molecular-based methods for the identification of *Candida auris.*

Assay	Identification From	Target	Reference
PCR and real-time qPCR (SYBR Green)	Colony	5.8S-ITS2-28S of rDNA	[31]
real-time qPCR (SYBR Green)	Swabs	5.8S-ITS2-28S	[32]
real-time qPCR (TaqMan)	Swabs and environmental sample	ITS2-rDNA	[33]
PCR	Colony	ITS1-5.8S-ITS2	[34]
Duplex PCR	Colony	GPI protein-encoding genes	[35]
Tetraplex PCR	Colony	26s rDNA	[36]
real-time qPCR (TaqMan)	Swabs	ITS2-rDNA	[37]
Multiplex end-point PCR	Colony	ITS 1-5.8S-ITS2	[38]
YEAST PANEL multiplex PCR	Colony and spiked serum samples	26S rDNA	[39]
GPS MONODOSE dtec-qPCR kit	Colony	Species-specific primers and probes	[39]
T2 Magnetic Resonance (T2MR) system	Swabs	Species-specific primers and probes	[41]
Loop-mediated isothermal amplification (LAMP)	Colony, swab and environmental sample	The ferredoxin oxidoreductase encoding gene	[42]

**Table 3 antibiotics-09-00778-t003:** Strengths and limitations of AFST used for *C. auris.*

Methods	Strengths and Limitations	Reference
Broth Microdilution Methods/Sensititre YeastOne	Determined ECVs are valuable in the analysis of MICs of isolates from the South Asian clade. MIC distributions can vary substantially for *C. auris* isolates from different clades. Are easy to perform.	[20]
*E*-test gradient diffusion method	Difficulty of interpretation for presence of aggregate directly adjacent to the zone of growth inhibition. The aggregates are present for evaluation of fluconazole, voriconazole, and anidulafungin but not in experiments performed with flucytosine or amphotericin B.	[20]
VITEK 2	MIC distributions can vary substantially for *C. auris* isolates from different clades	[28]
MBT ASTRA	MBT ASTRA has a potential to detect echinocandin nonsusceptible *C. auris* isolates within 6 h.	[57]
Molecular methods	Echinocandin resistance is mediated through limited mutations S639P or S639F in *FKS1*, and azole resistance through F126L, Y132F, and K143R in *ERG11* *	[2,5]

* To date, these are the only mutations associated with clinical failures due to azole and echinocandin drugs.

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
