# Peer review of "Candida auris: An Overview of How to Screen, Detect, Test and Control This Emerging Pathogen"

_antibiotics, 2020, doi:10.3390/antibiotics9110778_

Round 1

Reviewer 1 Report

In the present review, the authors provide an updated report on available methods for screening, identification and antifungal susceptibility of Candida auris with an emphasis on screening and control measures. It is particularly important for clinical laboratories to have proper indications on C. auris, an emerging pathogen associated with nosocomial outbreaks and difficult to correctly identify

The manuscript is clearly written, well organized and scientifically sound

Minor comments

  • Authors should summarize in the table the antifungal susceptibility tests and their respective strengths and limitations, so that readers can get an overview
  • Authors should check the style of references

Author Response

Dear Editor,

I would like to thank you for the opportunity of editing the manuscript for Antibiotics.

My co-authors and I apologize being previously unclear and we hope that the changes we made meet the reviewers’ demands.

In what follows we provide the list of the changes we made (responses are marked as “R” while the changes are interline in yellow in the text)

Reviewer 1

Authors should summarize in the table the antifungal susceptibility tests and their respective strengths and limitations, so that readers can get an overview

  1. Done

Authors should check the style of references

  1. Done. All the reference now meet the journal criteria i.e.: Author 1, A.B.; Author 2, C.D. Title of the article. Abbreviated Journal Name Year, Volume, page range.

Reviewer 2 Report

The review is a nice compilation of the methods that have been described to identify C. auris. It's well written although I found some typos (Authors line 243) and other minor errors like the name of the genes not italicized (FKS1, FKS2, ERG11) and Candida species not being italicized either. Also, I would add MS after mass spectrometry (line 211).

Author Response

Dear Reviewer ,

I would like to thank you for the opportunity of editing the manuscript for Antibiotics.

My co-authors and I apologize being previously unclear and we hope that the changes we made meet the reviewers’ demands.

In what follows we provide the list of the changes we made (responses are marked as “R” while the changes are interline in yellow in the text)

The review is a nice compilation of the methods that have been described to identify C. auris. It's well written although I found some typos (Authors line 243) and other minor errors like the name of the genes not italicized (FKS1FKS2ERG11) and Candida species not being italicized either. Also, I would add MS after mass spectrometry (line 211).

R. Done

Reviewer 3 Report

The manuscript "Candida auris: an overview of how to screen, detect, 2 test and control this emerging pathogen" presents an interesting and helpful review on identification of this specific Candida species and testing of resistance. There are only some minor spell checking (e.g. line 260: around 40% or =>of C. auris) and change in wording (e.g.. line 78 48 hours should be observed prior to sampling). In general cplease change hours to "working hours" in practice most tests need a longer duration of time.

Author Response

Dear Reviewer ,

I would like to thank you for the opportunity of editing the manuscript for Antibiotics.

My co-authors and I apologize being previously unclear and we hope that the changes we made meet the reviewers’ demands.

In what follows we provide the list of the changes we made (responses are marked as “R” while the changes are interline in yellow in the text)

The manuscript "Candida auris: an overview of how to screen, detect, 2 test and control this emerging pathogen" presents an interesting and helpful review on identification of this specific Candida species and testing of resistance. There are only some minor spell checking (e.g. line 260: around 40% or =>of C. auris) and change in wording (e.g.. line 78 48 hours should be observed prior to sampling). In general please change hours to "working hours" in practice most tests need a longer duration of time.

  1. Done

Reviewer 4 Report

Review 
Candida auris: an overview of how to screen, detect,  test and control this emerging pathogen.

There are many reviews currently published for C.auris. This text is quite nice and most probably up to date.

The text should also refer the most important related recent secondary literature.

A graphical take home message of what is best to "screen, detect,  test and control" would be of advantage.

Line 184:  clinical sensitivity (detection limits in the range of 1–10 CFU/reaction) and a specificity around 90% - please clarify:  90% specificity would be quite low. In 1 out of 10 C.auris is misdiagnosed by these methods ?

Author Response

Dear Reviewer ,

I would like to thank you for the opportunity of editing the manuscript for Antibiotics.

My co-authors and I apologize being previously unclear and we hope that the changes we made meet the reviewers’ demands.

In what follows we provide the list of the changes we made (responses are marked as “R” while the changes are interline in yellow in the text)

Candida auris: an overview of how to screen, detect,  test and control this emerging pathogen.

There are many reviews currently published for C.auris. This text is quite nice and most probably up to date.

The text should also refer the most important related recent secondary literature.

A graphical take home message of what is best to "screen, detect,  test and control" would be of advantage.

  1. Done

Line 184:  clinical sensitivity (detection limits in the range of 1–10 CFU/reaction) and a specificity around 90% - please clarify:  90% specificity would be quite low. In 1 out of 10 C.auris is misdiagnosed by these methods ?

  1. Yes, unfortunately C. auris could still be misdiagnosed by these methods.
